# Prior-Free Dynamic Auctions with Low Regret Buyers

**Yuan Deng**
Duke University
ericdy@cs.duke.edu

**Jon Schneider**
Google Research
jschnei@google.com

**Balasubramanian Sivan**
Google Research
balusivan@google.com

## Abstract

We study the problem of how to repeatedly sell to a buyer running a no-regret, mean-based algorithm. Previous work [Braverman et al., 2018] shows that it is possible to design effective mechanisms in such a setting that extract almost all of the economic surplus, but these mechanisms require the buyer's values each round to be drawn independently and identically from a fixed distribution. In this work, we do away with this assumption and consider the *prior-free setting* where the buyer's value each round is chosen adversarially (possibly adaptively).

We show that even in this prior-free setting, it is possible to extract a $(1 - \varepsilon)$-approximation of the full economic surplus for any $\varepsilon > 0$. The number of options offered to a buyer in any round scales independently of the number of rounds $T$ and polynomially in $\varepsilon$. We show that this is optimal up to a polynomial factor; any mechanism achieving this approximation factor, even when values are drawn stochastically, requires at least $\Omega(1/\varepsilon)$ options. Finally, we examine what is possible when we constrain our mechanism to a natural auction format where overbidding is dominated. Braverman et al. [2018] show that even when values are drawn from a known stochastic distribution supported on $[1/H, 1]$, it is impossible in general to extract more than $O(\log \log H / \log H)$ of the economic surplus. We show how to achieve the same approximation factor in the *prior-independent* setting (where the distribution is unknown to the seller), and an approximation factor of $O(1/\log H)$ in the *prior-free setting* (where the values are chosen adversarially).

## 1 Introduction

Revenue optimal auction design in settings where a seller interacts repeatedly with a buyer (like in the sale of Internet ads) is a problem of high commercial relevance. The promise of *dynamic auctions*, that allow the linking of buyers' decisions across time, is the significantly higher revenue they can achieve over running independent/decoupled auctions across time. The technical challenges that dynamic auctions introduce, along with their practical impact has inspired a lot of recent work in this area [Papadimitriou et al., 2016, Ashlagi et al., 2016, Mirrokni et al., 2018].

Traditionally, almost all work in dynamic mechanism design operates in the regime where the players' types (e.g. bidders' values) are drawn stochastically from a fixed distribution. In many situations this is far from a realistic assumption – for example, if the values of a buyer are modelled as a distribution, this underlying distribution likely drifts over time and is also subject to shocks determined by uncontrolled exogenous events. But this assumption is also in many ways critical: in a dynamic mechanism in an adversarial setting, a fully rational buyer (who cares about the effect of his current action on his future utility) would be unable to compute his future utility at any point of time in the game and thus unable to meaningfully best-respond.

On the other hand, auctions for digital ads have become increasingly more complex over time. The design space of dynamic auctions, in which a buyer bids on many items over the course of many

rounds, is very rich and has room for exceedingly complex auctions. A bidder may have difficulty behaving fully rationally in such an auction: the bidder may not have accurate priors for bidders, the bidder may not completely understand the mechanism, and finding an equilibrium might be computationally hard. Instead of acting fully rationally, a bidder might instead choose to try to learn how to bid over time, for example by using a no-regret learning algorithm. Recently, several streams of work (e.g. Agrawal et al. [2018], Braverman et al. [2018]) have explored the problem of how to design dynamic auctions for such bidders. In all cases these works assume, as is standard, that bidders' values are stochastically generated. However, one intriguing feature of modelling a bidder as a learning agent is that it no longer restricts us to the stochastic setting – the actions taken by a learning algorithm are perfectly well-defined in (and ostensibly even designed for) the *prior-free* setting where values are drawn adversarially. This opens a wealth of questions of how to robustly design dynamic mechanisms that perform well in the worst-case against some class of learning agents. In this paper, we explore this question for one of the simplest problems in dynamic mechanism design: repeatedly selling a single item to a single buyer for $T$ rounds.

We build off the setting of [Braverman et al., 2018], where they model the buyer as a learner running a mean-based low-regret algorithm. Intuitively, mean-based algorithms prefer to select actions that have performed historically well on average (it can be shown that many classic learning algorithms, like EXP3, Multiplicative Weights, and Follow-the-Perturbed-Leader, are all mean-based low-regret algorithms). In [Braverman et al., 2018], the authors show that surprisingly, when the buyer's values $v_t \in [0, 1]$ are drawn from a fixed distribution, it is possible to design a simple mechanism that obtains almost the full economic surplus (i.e., $\mathsf{Val} = \mathbb{E}[\sum_t v_t]$) as revenue. Their mechanism, however, relies crucially on the fact that the buyer's values are drawn from the same distribution every round. In particular, it is straightforward to verify that there exist sequences of values for the buyer that result in this mechanism receiving asymptotically zero total revenue.

In this paper we design mechanisms for this problem in the *prior-free* setting, when the buyer's values $v_t \in [0, 1]$ are chosen adversarially (possibly adaptively). In the course of doing this, we aim to minimize the complexity of our mechanisms, measured in terms of the number of distinct options (i.e. "bids") the mechanism presents to the bidder in any round. We call this quantity the option-complexity of the mechanism. Note that in mechanisms with high option-complexity it becomes harder to learn how to bid. If the option-complexity of the mechanism begins to scale with the number of rounds $T$, this may even nullify any sort of low-regret or mean-based guarantee the learning algorithm has (it may not even be possible to explore all potential options).

**Upper bound in the adversarial setting:** We design a non-adaptive (i.e., does not use historical bids/allocation/prices) *option-based mechanism* that yields a revenue of $\mathsf{Val} - O(\varepsilon T)$ with $O\left(\frac{\ln(1/\varepsilon)}{\varepsilon^3}\right)$ options, where the instance $(v_1, \cdots, v_T)$ is chosen by a (possibly adaptive) adversary and $\mathsf{Val}$ is the total economic surplus defined by $\mathsf{Val} = \sum_{t=1}^{T} v_t$.

**Lower bound in the stochastic (and hence adversarial) setting:** We show that even if values are drawn from an unknown stochastic distribution (i.e. in every round the buyer's value was drawn independently from some distribution $\mathcal{D}$), any non-adaptive option-based mechanism needs to offer at least $\Omega(1/\varepsilon)$ options to attain a $\mathsf{Val} - O(\varepsilon T)$ revenue. This implies the option-complexity of our algorithm is tight up to a polynomial factor in $1/\varepsilon$.

**Upper bound in the stochastic setting with unknown distribution via critical mechanisms:** Finally, although our mechanisms have relatively low option-complexity, they can still appear unnatural and complex. We examine what is possible by further restricting our mechanisms to *critical mechanisms* [Braverman et al., 2018], by imposing the desiderata of individual rationality, monotonicity of price and allocation in bid, and overbidding being dominated (see Section 2). Braverman et al. [2018] show that the seller can use a critical mechanism to extract a good revenue but not all of surplus, in particular showing the seller can always guarantee revenue equal to an $O(\frac{\log \log H}{\log H})$ fraction of total economic surplus when buyer values lie in the interval $\left[\frac{1}{H}, 1\right]$, and that this competitive ratio is tight. This critical mechanism requires full knowledge of the value distribution $\mathcal{D}$. We design a critical mechanism that achieves this same approximation factor, but in a *prior-independent setting where the distribution $\mathcal{D}$ is unknown*. In addition, we show that it is possible to achieve a slightly

worse competitive ratio of $O(\frac{1}{\log H})$ in the *prior-free* (adversarial values) setting by adapting existing prior-free mechanisms for the single-shot instance of this problem.

We emphasize that all the mechanisms we present are non-adaptive (i.e. allocation and payment rules at all times are fixed starting at the beginning of the protocol, and are not functions of the historical bids/allocations/payments) as in [Braverman et al., 2018].

## 1.1 Related Work

Our work is closely related to the dynamic mechanism design literature, such as [Balseiro et al., 2017, Liu and Psomas, 2017, Agrawal et al., 2018, Mirrokni et al., 2018, Balseiro et al., 2019], which studies how to sell items online to a fixed set of strategic buyers, whose valuations are fixed or drawn from some distributions. However, the buyers are fully strategic such that their bidding strategies aim to maximize their accumulative utility throughout the auction.

No-regret algorithms were first introduced in the context of the multi-armed bandit problem and have been widely studied (see Bubeck et al. [2012] for a survey). Applications of low-regret learning to algorithmic game theory are widespread (e.g. [Roughgarden, 2012, Syrgkanis and Tardos, 2013, Nekipelov et al., 2015, Daskalakis and Syrgkanis, 2016]). Most applications to dynamic auction design are from the perspective of seller attempting to learn the optimal auction against strategic buyers [Amin et al., 2013, 2014, Cole and Roughgarden, 2014, Morgenstern and Roughgarden, 2015, Devanur et al., 2016, Morgenstern and Roughgarden, 2016, Gonczarowski and Nisan, 2017, Cai and Daskalakis, 2017, Dudík et al., 2017, Drutsa, 2017, 2018, Liu et al., 2018]. Recent study takes the perspective of buyers and applies learning algorithms to help them learn how to bid in repeated and dynamic auctions [Feng et al., 2018, Balseiro et al., 2018].

In contrast to these works, we take the perspective of the sellers to design online auctions against the buyers who are running no-regret algorithms in bidding. As pointed out in a seminal empirical work [Nekipelov et al., 2015], bidders' behavior on Bing is largely consistent with a no-regret learning algorithm, which motivates a question of designing a dynamic mechanism against such a no-regret learning behavior. Braverman et al. [2018] initiated the study of mechanism design against a no-regret buyer when the buyer's valuations are drawn from a fixed and known distribution. In contrast to their works, we design mechanisms against a no-regret buyer in a prior-free / prior-independent setting.

## 2 Model and Preliminaries

Our setting is similar to the setting considered in [Braverman et al., 2018]: we consider a multiple round auction where every round a seller attempts to sell an item to a buyer running a low-regret (in fact, mean-based) algorithm to learn how to bid.

Specifically, we consider a $T$-round auction with one buyer and one seller. In each round $t$, there is one item for sale. At the beginning of this round, the buyer learns his private valuation $v_t \in \mathcal{V} \subseteq [0, 1]$ for this item. These valuations $v_t$ can be generated in one of two ways: (1) *Adversarial*, where $v_t$ is chosen arbitrarily by a (possibly adaptive) adversary; and (2) *Stochastic*, where $v_t$ is independently drawn from some distribution $\mathcal{D}$. This distribution $\mathcal{D}$ may either be known to the seller or not and we will mostly consider the case where $\mathcal{D}$ is unknown to the seller (i.e., the prior-independent setting).

For simplicity, we assume the values $v_t$ belong to a finite set $\mathcal{V}$. This is solely for the purpose of providing a finite number of different contexts to the buyer's learning algorithm and otherwise does not affect our mechanism at all.

To measure the performance of our mechanisms, we compare the revenue extracted from the mechanism to the *welfare*, the total value the buyer assigns to all the items.

**Definition 1.** *The welfare* $\mathsf{Val}(v_1, \cdots, v_T)$ *is equal to* $\sum_{t=1}^{T} v_t$.

The welfare clearly provides an upper bound on the revenue of our mechanisms. In cases where $v_t$ is drawn from some distribution $\mathcal{D}$, we will write $\mathsf{Val}(\mathcal{D}) = \mathbb{E}_{x \sim \mathcal{D}}[x] \cdot T$ to denote the expected welfare under this distribution.

## 2.1 Mechanism format

Since the buyer is running a learning algorithm, it is especially important to specify the manner of interaction between the buyer and the seller. We consider two classes of mechanisms for the seller: *option-based mechanisms*, and *critical mechanisms*.

In a option-based mechanism, the seller offers the buyer $K$ options (labeled 1 through $K$) each round. If the buyer selects choice $i$ at time $t$, the buyer receives the item with probability $a_{i,t}$ and pays a price $p_{i,t}$. A natural measure of complexity for such mechanisms is the number of options $K$ presented to the buyer, which we refer to as the *option-complexity* of the mechanism. Limiting this complexity is especially important when interacting with learning agents, as they require some time to explore each option (indeed, as $K$ approaches $T$, the low regret guarantees of the learning algorithms we consider become vacuous).

Critical mechanisms [Braverman et al., 2018] are a subset of option-based mechanisms that are *reasonable*. In a critical mechanism, the buyer interacts with the mechanism each round by submitting a bid $b$. The buyer then receives the item with probability $a_t(b)$ and pays a price $p_t(b)$. These allocation/payment rules should satisfy the following properties:

- *Individual rationality*: $p_t(b)$ satisfies $p_t(b) \leq b \cdot a_t(b)$, i.e. a bidder should never be charged more than their bid in expectation.
- *Monotonicity*: $p_t(b)$ and $a_t(b)$ are weakly increasing in $b$, i.e., submitting a higher bid should never decrease the winning probability or the payment.
- *Overbidding is dominated*: If the bidder's value is $v$, it should never be in their interest to submit a bid $b > v$, i.e. if $b > v$ then $v \cdot a_t(v) - p_t(v) > v \cdot a_t(b) - p_t(b)$ for all $t$.

In both option-based mechanisms and critical mechanisms, we assume that the seller is completely non-adaptive and sets the allocation / payment functions at the beginning of the protocol.

## 2.2 No-regret learner

In contrast to a utility-maximizing buyer, we consider a buyer who follows some no-regret strategy for the multi-armed bandit problem. In a classic multi-armed bandit problem with $T$ rounds, the learner (in our setting, the buyer) selects one of $K$ options ('arms') on round $t$ and receives a reward $r_{i,t} \in [0,1]$ if he selects option $i$. The rewards can be chosen adversarially and the learner's objective is to maximize his total reward.

Let $i_t$ be the arm pulled by the learner at round $t$. The *regret* for a (possibly randomized) strategy $\mathcal{A}$ is defined as the difference between performance of the strategy $\mathcal{A}$ and the best arm: $\mathsf{Reg}(\mathcal{A}) = \max_i \sum_{t=1}^T r_{i,t} - r_{i_t,t}$. A strategy $\mathcal{A}$ for the multi-armed bandit problem is *no-regret* if the expected regret is sub-linear in $T$, i.e., $\mathbb{E}[\mathsf{Reg}(\mathcal{A})] = o(T)$. In addition to the *bandits* setting in which the learner only learns the reward of the arm he pulls, our results also apply to the *experts* setting in which the learner can learn the rewards of all arms for every round. In our setting, the buyer learns $a_{i,t}$ and $p_{i,t}$, allowing him to compute the reward as $r_{i,t} = a_{i,t} \cdot v_t - p_{i,t}$. Moreover, the buyer has the additional information of her value $v_t$, and thus is in fact facing a *contextual bandit* problem.

**Contextual Bandits**  In a contextual bandit problem, the learner is additionally provided a *context* $c_t$ from a finite set $\mathcal{C}$. The reward of pulling arm $i$ under context $c$ on round $t$ is now given by $r_{i,t}(c)$. In the experts setting, the learner can obtain the values of $r_{i,t}(c_t)$ for all arms $i$ under context $c_t$ after round $t$, while the learner only learns $r_{i,t}(c_t)$ for the arm $i$ he pulls in the bandits setting.

The notion of regret for a strategy $\mathcal{M}$ can be easily extended to the contextual bandit problem by considering the best context-specific policy $\pi$: $\mathsf{Reg}(\mathcal{M}) = \max_{\pi:\mathcal{C}\to[K]} \sum_{t=1}^T r_{\pi(c_t),t}(c_t) - r_{i_t,t}(c_t)$. As before, a strategy $\mathcal{M}$ is no-regret if $\mathbb{E}[\mathsf{Reg}(\mathcal{M})] = o(T)$. When the size of the context $\mathcal{C}$ is a constant with respect to $T$, a no-regret strategy $\mathcal{M}$ for the contextual bandits can be simply constructed from a no-regret strategy $\mathcal{A}$ for the classic bandit problem: maintain a separate instance of $\mathcal{A}$ for every context $c \in C$ [Bubeck et al., 2012].

Among no-regret strategies, we are interested in a special class of *mean-based* strategies:

**Definition 2** (Mean-based Strategy)**.** *Let* $\sigma_{i,t}(c) = \sum_{s=1}^t r_{i,s}(c)$ *be the cumulative rewards for pulling arm $i$ under context $c$ for the first $t$ rounds. A strategy is $\gamma$-mean-based if whenever* $\sigma_{i,t}(c_t) <$

$\sigma_{j,t}(c_t) - \gamma T$, *the probability for the strategy to pull arm $i$ on round $t$ is at most $\gamma$. A strategy is mean-based if it is $\gamma$-mean-based with $\gamma = o(1)$.*

Intuitively, mean-based strategies are strategies that will pick the arm that historically performs the best. Braverman et al. [2018] shows that many no-regret algorithms are mean-based, including commonly used variants of EXP3 (for the bandits setting), the Multiplicative Weights algorithm (for the experts setting) and the Follow-the-Perturbed-Leader algorithm (for the experts setting).

## 3 Option-based Mechanisms

In this section, we demonstrate a mechanism that can extract full welfare from a mean-based no-regret learner even when the values are chosen adversarially.

### 3.1 Warm-up: Extracting Full Welfare for $\mathcal{V} = \{1, 2\}$

Consider an additive approximation target $\varepsilon > 0$. It is without loss of generality to consider the case with $2(1 - \varepsilon) > 1$: when $2(1 - \varepsilon) \leq 1$, the seller can simply implement a scheme with only one option that always allocates the item and charges a payment $2(1 - \varepsilon)$. We design a option-based mechanism with $K = \lceil \frac{\log \varepsilon}{\log(1-\varepsilon)} \rceil + 1$ choices in addition to the null choice in which the buyer receives and pays nothing for the entire time horizon. For the 0-th option, the buyer receives the item with probability $a_{0,t} = 1$ and pays a price $p_{0,t} = 2(1 - \varepsilon)$ for all $t$. As for the remaining $K - 1$ options, let $\kappa_i = \frac{\varepsilon}{(1-\varepsilon)^{i-1}} T$. We will divide the timeline of the $i$-th option with $1 \leq i \leq K$ into five sessions (see Table 1 for details).

For convenience, let $S_i = (\kappa_i, \kappa_{i+1}]$. Intuitively, the $i$-th option is *active* when $t \in S_i$, which spans $L_i = \kappa_{i+1} - \kappa_i = \frac{\varepsilon^2}{(1-\varepsilon)^i} T$ rounds. Among these $L_i$ rounds, the item is always allocated to the buyer with probability 1 while the payment changes in a way such that: the payment for the first $\varepsilon L_i$ rounds is 0, the payment for the last $\varepsilon L_i$ rounds is 2, and the payment for the remaining rounds is 1.

| Session | Start Time | End Time | Allocation Prob. | Payment |
|---------|-----------|----------|------------------|---------|
| $\emptyset_1$ | 0 | $\kappa_i$ | 0 | 0 |
| 0 | $\kappa_i$ | $\kappa_i + \frac{\varepsilon^3}{(1-\varepsilon)^i} T$ | 1 | 0 |
| 1 | $\kappa_i + \frac{\varepsilon^3}{(1-\varepsilon)^i} T$ | $\kappa_{i+1} - \frac{\varepsilon^3}{(1-\varepsilon)^i} T$ | 1 | 1 |
| 2 | $\kappa_{i+1} - \frac{\varepsilon^3}{(1-\varepsilon)^i} T$ | $\kappa_{i+1}$ | 1 | 2 |
| $\emptyset_2$ | $\kappa_{i+1}$ | $T$ | 0 | 0 |

Table 1: Construction of the $i$-th option

Assume the buyer is running a $\gamma$-mean-based algorithm. To analyze the revenue guarantee of our mechanism, we consider an arbitrary sequence of valuations $(v_1, \cdots, v_T)$ and $\mathsf{Val} = \sum_t v_t$. The high level idea behind this construction is that for the high valuations, i.e, $v_t = 2$, the utility $\sigma_{i,t}(2)$ keeps increasing as $t$ increases for *the high option* ($i = 0$) while for *the low options* ($i > 0$), it only increases within the active period $S_i$. Therefore, with sufficiently large $t$, we have $\sigma_{0,t}(2) > \sigma_{i,t}(2)$ for all $i > 0$ and therefore, the buyer with high valuation will play the high option with high probability. As for $v_t = 1$, the buyer plays the high option with probability at most $\gamma$ since its payment is too high and we argue that the buyer will play the $i$-th option with high probability when $t \in S_i$.

**High valuation** Assume that $v_t = 2$. First notice that the cumulative utility for playing the 0-th option is $\sigma_{0,t}(2) = \varepsilon t \cdot 2$. Suppose $t \in S_{i*}$ for some $i^*$. For $i < i^*$, the active period of the $i$-th option with $i < i^*$ is already past and the cumulative utility for playing the $i$-th option is at most

$$\sigma_{i,t}(2) \leq L_i \cdot 2 = \frac{\varepsilon^2}{(1-\varepsilon)^i} T \cdot 2 \leq \frac{\varepsilon^2}{(1-\varepsilon)^{i^*-1}} T \cdot 2 = \varepsilon \cdot \kappa_{i^*} \cdot 2 = \sigma_{0,t}(2) - \varepsilon \cdot (t - \kappa_{i^*}) \cdot 2$$

As for the $i^*$-th option, we have $\sigma_{i^*,t}(2) \leq (t - \kappa_{i^*}) \cdot 2 = \sigma_{0,t}(2) - (\kappa_{i^*} - (1 - \varepsilon)t) \cdot 2$.

Moreover, for any $i$-th option with $i > i^*$, we simply have $\sigma_{i,t}(2) = 0$. Therefore, the buyer with valuation $v_t = 2$ for $t \in S_{i^*}$ will play the 0-th option with probability at least $1 - K\gamma$

when $\varepsilon t \cdot 2 > \gamma T$, $\varepsilon \cdot (t - \kappa_{i^*}) \cdot 2 > \gamma T$, and $(\kappa_{i^*} - (1-\varepsilon)t) \cdot 2 > \gamma T$, which implies that $\kappa_{i^*} + \frac{\gamma}{2\varepsilon} \cdot T < t < \kappa_{i^*+1} - \frac{\gamma}{2(1-\varepsilon)} \cdot T$. Therefore, for each time period $S_i$ with $1 \le i \le K$, there are at least $L_i - \left(\frac{\gamma}{2\varepsilon} + \frac{\gamma}{2(1-\varepsilon)}\right) T$ rounds where the buyer has probability at least $1 - K\gamma$ to play the 0-th option, which contributes $2(1-\varepsilon)$ revenue per round. Therefore, the expected revenue loss from time period $S_i$ is at most

$$2\varepsilon \cdot L_i + \left(\frac{\gamma}{2\varepsilon} + \frac{\gamma}{2(1-\varepsilon)}\right) T \cdot 2 + K\gamma \cdot L_i \cdot 2$$

where $2\varepsilon \cdot L_i$ is the revenue loss of charging $2(1-\varepsilon)$ and $K\gamma \cdot L_i \cdot 2$ is the expected revenue loss from playing an option other than the 0-th option. Thus, the total expected revenue loss from the rounds when $v_t = 2$ is at most

$$(\varepsilon T) \cdot 2 + \sum_i \left[ 2\varepsilon \cdot L_i + \left(\frac{\gamma}{2\varepsilon} + \frac{\gamma}{2(1-\varepsilon)}\right) T \cdot 2 + K\gamma \cdot L_i \cdot 2 \right] = O(\varepsilon T)$$

where $(\varepsilon T) \cdot 2$ is the revenue loss from the first $\varepsilon T$ rounds.

**Low valuation**  Assume that $v_t = 1$. First notice that after the first $\varepsilon T$ rounds, the cumulative utility for playing the 0-th option is $\sigma_{0,t}(1) = (1 - 2(1-\varepsilon))t = -\Omega(T)$. Since there is a null arm that provides cumulative utility 0, the buyer's probability of playing the 0-th option is at most $\gamma$.

Suppose $t \in S_{i^*}$ for some $i^*$. From our construction of the $i$-th option for any $i \ne i^*$, the buyer's cumulative utility of playing the $i$-th option is exactly 0: the buyer's utility gain is 0 in from session $\emptyset_1, \emptyset_2$, and 1, while her utility gain from session 0 is exactly cancelled out with his utility loss from session 2, which leads to $\sigma_{i,t}(1) = 0$ for $t > \kappa_{i+1}$ or $t < \kappa_i$. As for the $i^*$-th option, we have

$$\sigma_{i^*,t}(1) = \begin{cases} t - \kappa_i & \text{for } t \text{ in session } 0 \\ \frac{\varepsilon^3}{(1-\varepsilon)^{i^*}}T & \text{for } t \text{ in session } 1 \\ \kappa_{i+1} - t & \text{for } t \text{ in session } 2 \end{cases}$$

Therefore, once $\kappa_i + \gamma T < t < \kappa_{i+1} - \gamma T$, the buyer with $v_t = 1$ will play the $i^*$-th option with probability $1 - K\gamma$. Therefore, the expected revenue loss within the time period $S_i$ is $2\gamma T + K\gamma \cdot L_i$, where $K\gamma \cdot L_i$ is the expected revenue loss from playing an option other than the $i^*$-th option. Thus, the total revenue loss from the rounds with $v_t = 1$ is at most $\varepsilon T + \sum_{i=1}^K K\gamma \cdot L_i = O(\varepsilon T)$ where $\varepsilon T$ is the revenue loss from the first $\varepsilon T$ rounds.

## 3.2  Extracting Full Welfare for $\mathcal{V} = \{1, \cdots, H\}$

We provide an option-based mechanism with $K = H \cdot \lceil \frac{3H^2}{\varepsilon} \rceil$ options that achieves an additive revenue loss $O(\ln H \cdot \varepsilon T)$ for $\mathcal{V} = \{1, \cdots, H\}$. As usual, we assume that there is always a null choice in which the buyer receives and pays nothing for the entire time horizon. For convenience, let $G_i = \sum_{\tau=1}^i \frac{1}{\tau}$ be the sum of the harmonic series up to $i$ and $\alpha = 1 - \frac{1}{3H}$. Moreover, $\kappa_{i,j} = (G_H + 2\alpha) \cdot \frac{\varepsilon T}{H} + (j-1) \cdot \frac{\varepsilon T}{3H^2}$ where $i \in \mathcal{V}$ and $1 \le j \le \lceil \frac{3H^2}{\varepsilon} \rceil$. Although $\kappa_{i,j}$ only depends on $j$, we still use the notation $\kappa_{i,j}$ for clarity. We will divide the timeline of the $(i,j)$-th option into five sessions (see Table 2).

| Session | Start Time | End Time | Allocation Prob. | Payment |
|---------|-----------|----------|------------------|---------|
| $init$ | 0 | $\alpha \cdot \frac{\varepsilon T}{H}$ | 0 | $i$ |
| 0 | $\alpha \cdot \frac{\varepsilon T}{H}$ | $\kappa_{i,j} - (G_i + \alpha) \cdot \frac{\varepsilon T}{H}$ | 0 | 0 |
| $ready$ | $\kappa_{i,j} - (G_i + \alpha) \cdot \frac{\varepsilon T}{H}$ | $\kappa_{i,j}$ | 1 | 0 |
| 1 | $\kappa_{i,j}$ | $\kappa_{i,j+1}$ | 1 | $i$ |
| $\emptyset$ | $\kappa_{i,j+1}$ | $T$ | 0 | $H$ |

Table 2: Construction of the $(i,j)$-th option

Assume the buyer is running a $\gamma$-mean-based algorithm. To analyze the revenue guarantee of our mechanisms, we consider an arbitrary sequence of valuations $(v_1, \cdots, v_T)$ and $\mathsf{Val} = \sum_t v_t$.

Intuitively, the $(i,j)$-th option starts with a $init$ session in which it does not allocate the item but charges a payment $i$, followed by a 0 session in which the option allocates and charges nothing.

Therefore, the buyer will not play the $(i, j)$-th option before its $ready$ session. In the $ready$ session, the option allocates the item for free while in the $1$ session, the option allocates the item with a payment $i$. Our construction ensures that if $v_t = i$ for $t \in (\kappa_{i,j}, \kappa_{i,j+1}]$, then the buyer will play the option $(i, j)$ with high probability, which generates revenue $i$.

**Lemma 3.** *If* $t \in (\kappa_{i,j} + \gamma T, \kappa_{i,j+1} - \gamma T]$, *then for any option* $(i', j')$ *with* $i' \neq i$ *or* $j' \neq j$, $\sigma_{(i,j),t}(i) - \sigma_{(i',j'),t}(i) > \gamma T$.

Therefore, for $v_t = i$ with $t \in (\kappa_{i,j} + \gamma T, \kappa_{i,j+1} - \gamma T]$, the buyer will play option $(i, j)$ with probability at least $1 - K\gamma$, which generates revenue $i$ per round. Thus, the revenue loss is at most

$$H \cdot (G_H + 2\alpha) \cdot \frac{\varepsilon T}{H} + H \cdot 2\gamma T \cdot K + K\gamma \cdot H \cdot T = O(\ln H \cdot \varepsilon T)$$

where $H \cdot (G_H + 2\alpha) \cdot \frac{\varepsilon T}{H}$ is the revenue loss for the first $\max_i \kappa_{i,1} = (G_H + 2\alpha) \cdot \frac{\varepsilon T}{H}$ rounds, $H \cdot 2\gamma T \cdot K$ is the revenue loss for $t \in (\kappa_{i,j}, \kappa_{i,j} + \gamma T]$ or $t \in (\kappa_{i,j+1} - \gamma T, \kappa_{i,j+1}]$, and $K\gamma \cdot H \cdot T$ is the revenue loss from playing an undesired option.

**Theorem 4.** *If the buyer with* $\mathcal{V} = \{1, 2, \cdots, H\}$ *is running a mean-based algorithm, for any constant* $\varepsilon > 0$, *there exists a non-adaptive option-based mechanism with* $O(\frac{H^3 \ln H}{\varepsilon})$ *options for the seller which obtains revenue at least* $\mathsf{Val} - O(\varepsilon T)$.

### 3.3 Extracting Full Welfare for $\mathcal{V} \subseteq [0, 1]$

Let $\varepsilon$ be parameter for the target additive revenue loss $O(\varepsilon T)$. For ease of presentation, we will rescale $\mathcal{V}$ to $[0, H]$ such that $H = 1/\varepsilon$, and thus, it suffices to show that we can obtain $O(T)$ loss in the scaled version. First notice that it suffices to consider $\mathcal{V} \subseteq [1, H]$ since for all valuations less than $1$, we will suffer revenue loss at most $1$ from each of them.

**Lemma 5.** *Consider* $v_t$ *such that* $i < v_t < i + 1$ *and* $t \in (\kappa_{i,j}, \kappa_{i,j+1}]$. *Then, for any option* $(i', j')$ *with* $i' \notin \{i, i+1\}$ *or* $j' > j$, $\max\{\sigma_{(i,j),t}(v_t), \sigma_{(i+1,j),t}(v_t)\} - \sigma_{(i',j'),t}(v_t) > \gamma T$.

Therefore, with probability at least $1 - K\gamma$, the buyer satisfying the requirement of Lemma 5 will play either option $(i, j')$ or option $(i + 1, j')$ with $j' \leq j$. Recall that it is in fact that $\kappa_{i,j} = \kappa_{i+1,j}$ for all $i$. Therefore, if the buyer plays option $(i + 1, j)$, it will generate revenue $i + 1$ since option $(i + 1, j)$ is also in its $1$ session. Moreover, if the buyer plays option $(i, j')$ or $(i + 1, j')$ with $j' < j$, then the option is already in its $\emptyset$ session and the buyer needs to pay $H$.

Thus, the revenue loss from $v_t$ is at most $1$. Applying a similar argument as in Section 3.2, we can conclude that the expected revenue loss is $O(T)$. Rescale it back to $\mathcal{V} = [0, 1]$, we have

**Theorem 6.** *If the buyer with* $\mathcal{V} \subseteq [0, 1]$ *is running a mean-based algorithm, for any constant* $\varepsilon > 0$, *there exists a non-adaptive option-based mechanism with* $O(\frac{\ln 1/\varepsilon}{\varepsilon^3})$ *options for the seller which obtains revenue at least* $\mathsf{Val} - O(\varepsilon T)$.

Meanwhile, we provide a lower-bound on the option-complexity, which implies the option-complexity of our algorithm is tight up to a polynomial factor in $\frac{1}{\varepsilon}$.

**Theorem 7.** *If the buyer with* $\mathcal{V} \subseteq [0, 1]$ *is running a mean-based algorithm, an option-based mechanism, which obtains expected revenue at least* $\mathsf{Val} - O(\varepsilon T)$, *must have* $\Omega(\frac{1}{\varepsilon})$ *options.*

*Proof.* We first prove a lower bound for $\mathcal{V} = \{1, 2, \cdots, H\}$ and the theorem will be a simple corollary of this lower bound. Let $I_{i,t}(c)$ be a binary variable indicating whether the buyer with value $v_t = c$ plays the $i$-th option. Suppose there are $K$ options in total and let $P_i(c) = \sum_{t=1}^{T} \Pr[I_{i,t}(c) = 1] \cdot p_{i,t}$ be the expected total revenue obtained from the $i$-th option when the buyer's valuations are $v_t = c$ for all $t$. Since the expected total revenue is at least $\mathsf{Val} - O(\varepsilon T)$, when the buyer's valuations are $v_t = 1$ for all $t$ in which the total expected revenue is at least $T - \mu\varepsilon T$ for some constant $\mu$, there must exist an option $i^*$ such that $P_{i^*}(1) \geq \frac{(1 - \mu\varepsilon)T}{K}$. Moreover, let $t^* = \sup\{t \mid \sigma_{i^*,t}(1) \geq -\gamma T\}$. $t^*$ is well-defined since $\sigma_{i^*,0}(1) = 0$. Notice that for all $t > t^*$, since the buyer is running a mean-based algorithm, we have $\Pr[I_{i^*,t}(1)] \leq \gamma$ due to the presence of the null option. Therefore, we have

$$\sum_{t \leq t^*} p_{i^*,t} + \sum_{t > t^*} \gamma \cdot p_{i^*,t} \geq P_{i^*}(1) \geq \frac{(1 - \mu\varepsilon)T}{K} \Rightarrow \sum_{t \leq t^*} p_{i^*,t} \geq \frac{(1 - \mu\varepsilon)T}{K} - \gamma H T.$$

where we use the fact that $0 \leq p_{i^*,t} \leq H$. Notice that the cumulative utility $\sigma_{i^*,t^*}(H)$ is

$$
\begin{aligned}
\sigma_{i^*,t^*}(H) &= \sum_{t \leq t^*} H \cdot a_{i^*,t} - p_{i^*,t} = H \cdot \sigma_{i^*,t^*}(1) + (H-1) \sum_{t \leq t^*} p_{i^*,t} \\
&\geq \frac{(H-1)(1-\mu\varepsilon)T}{K} - \gamma H^2 T
\end{aligned}
$$

Consider an environment when the buyer's valuations are $v_t = H$ for all $t$. Since the buyer is running a no-regret algorithm, her cumulative utility for the first $t^*$ rounds is at least $\sigma_{i^*,t^*}(H) - o(T)$. This is true because although the standard no-regret guarantee only applies to the final round $T$, the regret for the first $t$ rounds must also be $o(T)$, for any $t < T$. For the sake of contradiction, assume that the regret for the first $t$ rounds is $\Omega(T)$. Notice that the no-regret algorithm does not depend on the future. Therefore, consider an environment where the rewards for all options after round $t$ are set to be 0, which results in a $\Omega(T)$ regret for the final round $T$. A contradiction.

In addition, notice that the revenue loss from the first $t^*$ rounds is at least the buyer's cumulative utility, and thus, the revenue loss is at least $\sigma_{i^*,t^*}(H) - o(T) = \frac{(H-1)T}{K} - O(\varepsilon T)$. Finally, since the total revenue loss for $T$ rounds is at least the total revenue loss for the first $t^*$ rounds, in order to achieve $O(\varepsilon T)$ revenue loss, we must have $K = \Omega(\frac{H}{\varepsilon})$. $\qquad\square$

Observe that our proof only uses two sequences of valuations: a sequence with all 1 and a sequence of all $H$. Thus, our lower bound also applies to the stochastic settings with unknown distributions.

# 4 Critical mechanisms

In this section we examine what the seller can accomplish when restricted to a critical mechanism.

With option-based mechanisms, we have shown in the previous section that it is possible to extract arbitrarily close to the full welfare even when the buyer's values are chosen adversarially. In contrast to this, Braverman et al. [2018] show that with a critical mechanism, it is impossible to achieve even a constant-factor approximation to the buyer's welfare, even when the buyer's values are drawn from a distribution known to the seller.

**Theorem 8** (Corollary C.13 of [Braverman et al., 2018]). *Let $R(\mathcal{D})$ be the maximum possible revenue a seller using a non-adaptive critical mechanism can achieve when the buyer's values are drawn independently each round from distribution $\mathcal{D}$. Then the ratio $R(\mathcal{D})/\mathsf{Val}(\mathcal{D})$ can grow arbitrarily small. If $\mathcal{D}$ is supported on an interval $[1, H]$, then this ratio can be as small as $O(\log\log H/\log H)$.*

In Braverman et al. [2018], the authors also demonstrate how to construct a simple mechanism which achieves this maximum possible revenue (and hence this $O(\log\log H/\log H)$ competitive ratio to the welfare), but their construction requires detailed knowledge of the distribution $\mathcal{D}$.

## 4.1 Values from an unknown distribution

We show that it is possible achieve this same competitive ratio to the welfare in the prior-independent setting, where the seller does not know the distribution $\mathcal{D}$ but only a range $[1, H]$ it is supported on. In our mechanism, at each time $t$ the seller specifies a reserve price $f(t)$, where $f$ is a decreasing function with range $[1, H]$ such that $f(t) = \max\left(\exp\left(\frac{1}{C} \cdot (1 - \eta - \frac{t}{T})\right), 1\right)$, where $\eta = (1 + \log H)^{-\varepsilon}$ and $C = \frac{1-\eta}{1+\log H}$ for $\varepsilon \in (0, 1)$. In each round, if the buyer bids above $f(t)$ they receive the item and pay $b$; otherwise, they do not receive the item and pay nothing. More formally, the allocation and payment rules $(a_t(b), p_t(b))$ are defined as follows: if $b \geq f(t)$, then $p_t(b) = b$, and $a_t(b) = 1$; otherwise, $p_t(b) = a_t(b) = 0$.

**Theorem 9.** *There is a non-adaptive critical mechanism for the seller which obtains expected revenue at least $O(\log\log H/\log H)\mathsf{Val}(\mathcal{D})$ from any buyer running a mean-based algorithm whose values are drawn independently each round from some distribution $\mathcal{D}$ supported on $[1, H]$. This mechanism depends only on $H$ and not on $\mathcal{D}$.*

Consider the function $x(v) : [1, H] \to [0, T]$ where $x(v) = 1 - \frac{1}{T} \cdot \min_{f(t) \leq v} t$. Note that $x(v)$ equals the number of rounds where a bidder with value $v$ has value higher than the reserve price $f(t)$

(in particular, $x(v)$ is an increasing function of $v$). It can be shown (Braverman et al. [2018], Section C) that if the buyer is mean-based, the revenue obtained by the seller by using such an auction is given by $R(\mathcal{D}) = T \cdot \mathbb{E}_{v \sim \mathcal{D}} [vx(v) - \max_w(v - w)x(w)] - o(T)$.

**Lemma 10.** *If the seller is using a first-price auction with decreasing reserve price, then $R(\mathcal{D})/\mathsf{Val}(\mathcal{D})$ is maximized when $\mathcal{D}$ is a singleton distribution.*

*Proof of Theorem 9.* Note that for this choice of $f$, $x(v) = \eta + C \log v$. By Lemma 10, $R(\mathcal{D})/\mathsf{Val}(\mathcal{D})$ is maximized when $\mathcal{D}$ is a singleton distribution. We therefore have that:

$$
\begin{aligned}
\frac{R(\mathcal{D})}{\mathsf{Val}(\mathcal{D})T} &\geq \min_v \frac{vx(v) - \max_w(v - w)x(w)}{v} = \min_{v, w < v} \left( x(v) - \left(1 - \frac{w}{v}\right) x(w) \right) \\
&= \min_{v, w < v} \left( C \log \frac{v}{w} + \frac{w}{v}(\eta + C \log w) \right).
\end{aligned}
$$

For a fixed $w$, this is minimized when $v = w \left( \frac{\eta}{C} + \log w \right)$. It follows that

$$
\begin{aligned}
\frac{R(\mathcal{D})}{\mathsf{Val}(\mathcal{D})T} &\geq \min_w \left( C \log \left( \frac{\eta}{C} + \log w \right) + 1 \right) \geq C \log \left( \frac{\eta}{C} \right) \\
&\geq \frac{(1 - (1 + \log H)^{-\varepsilon}) \log((1 + \log H)^{1 - \varepsilon})}{1 + \log H} = \Theta \left( \frac{\log \log H}{\log H} \right).
\end{aligned}
$$

$\square$

### 4.2 Adversarial values

Additionally, when the buyer's values are drawn adversarially, we show that it is possible to achieve a slightly worse competitive ratio of $O(1/\log H)$. This follows naturally from the known fact that it is possible to achieve the same approximation guarantee against a *strategic* buyer with value in $[1, H]$ playing a single-round version of this game (see e.g. Chapter 6 of Hartline [2013]) – we simply show that if we run this mechanism every round, mean-based buyers will learn to bid in the same manner as strategic buyers. Our mechanism (equivalent to the mechanism presented in Theorem 6.5 of Hartline [2013]) is as follows: for each $b$ and for all $t$, we set the allocation probability $a_t(b) = (1 + \log b)/(1 + \log H)$ and the expected price charged to $p_t(b) = b/(1 + \log H)$. Note that this mechanism can also be interpreted as a second-price auction where the seller draws a random reserve from the distribution with a cumulative density function $F(r) = \frac{1 + \log r}{1 + \log H}$. It can be seen that for any $v \in [1, H]$, the expected utility $U(v, b) = v \cdot a_t(b) - p_t(b)$ of bidding $b$ with value $v$, is maximized when $b = v$. A strategic buyer therefore will always bid their value, and pay $1/(1 + \log H)$ of their value in total.

Intuitively, the mean-based guarantee ensures that a mean-based buyer will (most of the time) choose a bid close to $v$, and thus it contributes a similar amount of revenue as a strategic buyer.

**Theorem 11.** *There is a non-adaptive critical mechanism for the seller which obtains expected revenue at least $O(1/\log H)\mathsf{Val}$ from any buyer running a mean-based algorithm whose values are adversarially set but lie in the interval $[1, H]$. This mechanism depends only on $H$ and not on $\mathcal{D}$.*

## 5 Conclusion

In this work, we design mechanisms against a no-regret, mean-based buyer in prior-independent and prior-free setting. We show that using option-based mechanism can extract almost full welfare in a prior-free setting. For critical mechanisms, our mechanism in the prior-independent setting matches the best-known guarantee for the prior-dependent setting in the literature, and we obtain a slightly worse guarantee for the prior-free setting. A nature direction for future work is to understand what can be achieved in an environment with multiple learning buyers. Moreover, while both our works and [Braverman et al., 2018] focus on the revenue guarantee of the seller against a no-regret buyer, it is interesting to understand what kinds of the buyer's learning strategy can lead to a good utility performance. Furthermore, what combinations of the buyer's learning strategy and the seller's mechanism can achieve a socially desirable outcome?

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
