[Supplementary Material]

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

# Appendix

## A   Appendix for Option-based Mechanism

### A.1   Proof of Lemma 3

| Session | $\sigma_{(i,j),t}(v)$ |
|---------|----------------------|
| $init$ | $-t \cdot i$ |
| $0$ | $-\alpha \cdot \frac{\varepsilon T}{H} \cdot i$ |
| $ready$ | $-\alpha \cdot \frac{\varepsilon T}{H} \cdot i + \left(t - \kappa_{i,j} + (G_i + \alpha) \cdot \frac{\varepsilon T}{H}\right) \cdot v$ |
| $1$ | $-\alpha \cdot \frac{\varepsilon T}{H} \cdot i + (G_i + \alpha) \cdot \frac{\varepsilon T}{H} \cdot v + (t - \kappa_{i,j}) \cdot (v - i)$ |
| $\emptyset$ | $-\alpha \cdot \frac{\varepsilon T}{H} \cdot i + (G_i - \alpha) \cdot \frac{\varepsilon T}{H} \cdot v + \frac{\varepsilon T}{3H^2} \cdot (v - i) - (t - \kappa_{i,j+1})H$ |

Table 3: Cumulative utility of the $(i,j)$-th option

*Proof.* Notice that for a valuation $v_t = i$ with $t \in (\kappa_{i,j}, \kappa_{i,j+1}]$, its cumulative utility for playing option $(i,j)$ is

$$\sigma_{(i,j),t}(i) = -\alpha \cdot \frac{\varepsilon T}{H} \cdot i + (G_i + \alpha) \cdot \frac{\varepsilon T}{H} \cdot i + (t - \kappa_{i,j}) \cdot (i - i) = G_i \cdot \frac{\varepsilon T}{H} \cdot i. \tag{1}$$

$v_t$'s cumulative utility for playing option $(i', j')$ with $i' < i$ is at most

$$\begin{aligned}
\sigma_{(i',j'),t}(i) &\leq -\alpha \cdot \frac{\varepsilon T}{H} \cdot i' + (G_{i'} + \alpha) \cdot \frac{\varepsilon T}{H} \cdot i + (t - \kappa_{i',j'}) \cdot (i - i') \\
&\leq -\alpha \cdot \frac{\varepsilon T}{H} \cdot i' + (G_{i'} + \alpha) \cdot \frac{\varepsilon T}{H} \cdot i + \frac{\varepsilon T}{3H^2} \cdot (i - i') \\
&= G_{i'} \cdot \frac{\varepsilon T}{H} \cdot i + (\alpha + \frac{1}{3H}) \cdot \frac{\varepsilon T}{H} \cdot (i - i') \\
&= G_{i'} \cdot \frac{\varepsilon T}{H} \cdot i + \frac{\varepsilon T}{H} \cdot (i - i')
\end{aligned}$$

Taking the difference between $\sigma_{(i',j'),t}(i)$ and $\sigma_{(i,j),t}(i)$, we have

$$\begin{aligned}
\sigma_{(i,j),t}(i) - \sigma_{(i',j'),t}(i) &\geq (G_i - G_{i'}) \cdot \frac{\varepsilon T}{H} \cdot i - \frac{\varepsilon T}{H} \cdot (i - i') \\
&= \left(\sum_{\tau=i'+1}^{i} \frac{i}{\tau}\right) \cdot \frac{\varepsilon T}{H} - \frac{\varepsilon T}{H} \cdot (i - i') \\
&= \left(i - i' + \sum_{\tau=i'+1}^{i} \left(\frac{i}{\tau} - 1\right)\right) \cdot \frac{\varepsilon T}{H} - \frac{\varepsilon T}{H} \cdot (i - i') \\
&= \left(\sum_{\tau=i'+1}^{i} \left(\frac{i}{\tau} - 1\right)\right) \cdot \frac{\varepsilon T}{H} > \gamma T
\end{aligned}$$

In addition, $v_t$'s cumulative utility for playing option $(i', j')$ with $i' > i$ is at most

$$\sigma_{(i',j'),t}(i) \leq \max\left\{-\alpha \cdot \frac{\varepsilon T}{H} \cdot i' + (G_{i'} + \alpha) \cdot \frac{\varepsilon T}{H} \cdot i + (t - \kappa_{i',j'}) \cdot (i - i'), 0\right\}$$

If the maximum is taken by 0, then it is clear that $\sigma_{(i,j),t}(i) - \sigma_{(i',j'),t}(i) > \gamma T$. On the other hand,

$$\begin{aligned}
\sigma_{(i',j'),t}(i) &\leq -\alpha \cdot \frac{\varepsilon T}{H} \cdot i' + (G_{i'} + \alpha) \cdot \frac{\varepsilon T}{H} \cdot i + (t - \kappa_{i',j'}) \cdot (i - i') \\
&\leq -\alpha \cdot \frac{\varepsilon T}{H} \cdot i' + (G_{i'} + \alpha) \cdot \frac{\varepsilon T}{H} \cdot i \\
&= G_{i'} \cdot \frac{\varepsilon T}{H} \cdot i + \alpha \cdot \frac{\varepsilon T}{H} \cdot (i - i')
\end{aligned}$$

433   Taking the difference between $\sigma_{(i',j'),t}(i)$ and $\sigma_{(i,j),t}(i)$, we have

$$
\begin{aligned}
\sigma_{(i,j),t}(i) - \sigma_{(i',j'),t}(i) &\geq (G_i - G_{i'}) \cdot \frac{\varepsilon T}{H} \cdot i + \alpha \cdot \frac{\varepsilon T}{H} \cdot (i' - i) \\
&= \left( -\sum_{\tau=i+1}^{i'} \frac{i}{\tau} \right) \cdot \frac{\varepsilon T}{H} + \alpha \cdot \frac{\varepsilon T}{H} \cdot (i' - i) \\
&= \left( -(i' - i) + \sum_{\tau=i+1}^{i'} \frac{\tau - i}{\tau} \right) \cdot \frac{\varepsilon T}{H} + \alpha \cdot \frac{\varepsilon T}{H} \cdot (i' - i) \\
&\geq \left( -(i' - i) + \frac{i' - i}{H} \right) \cdot \frac{\varepsilon T}{H} + \alpha \cdot \frac{\varepsilon T}{H} \cdot (i' - i) \\
&\geq \frac{2}{3H} \cdot \frac{\varepsilon T}{H} \cdot (i' - i) > \gamma T
\end{aligned}
$$

434   Moreover, notice that $(\kappa_{i,j}, \kappa_{i,j+1}]$ are disjoint intervals for a fix $i$. In addition, for $\kappa_{i,j} + \gamma T < t <$
435   $\kappa_{i,j+1} - \gamma T$, we have for any $j' < j$,

$$
\sigma_{(i,j'),t}(i) = \sigma_{(i,j'),\kappa_{i,j'+1}}(i) - (t - \kappa_{i,j'+1})H \leq G_i \cdot \frac{\varepsilon T}{H} \cdot i - H \cdot \gamma T
$$

436   where we use the fact that $\sigma_{(i,j'),\kappa_{i,j'+1}}(i) = G_i \cdot \frac{\varepsilon T}{H} \cdot i$ for all $j'$ due to (1). Similarly, for $j' > j$,

$$
\sigma_{(i,j'),t}(i) = \sigma_{(i,j'),\kappa_{i,j'}}(i) - (\kappa_{i,j'} - t)i \leq G_i \cdot \frac{\varepsilon T}{H} \cdot i - i \cdot \gamma T
$$

437   where we use the fact that $\sigma_{(i,j'),\kappa_{i,j'}}(i) = G_i \cdot \frac{\varepsilon T}{H} \cdot i$ for all $j'$ due to (1). $\qquad\square$

## A.2   Analysis for Extracting Full Welfare for $\mathcal{V} \subseteq [0,1]$

439   Let $\varepsilon'$ be parameter for the target additive revenue loss $O(\varepsilon' T)$. For ease of presentation, we will
440   rescale $\mathcal{V}$ to $[0, H]$ such that $H = 1/\varepsilon'$, and thus, it suffices to show that we can obtain $O(T)$ loss in
441   the scaled version. First notice that it suffices to consider $\mathcal{V} = [1, H]$ since for all valuations less than
442   1, we will suffer revenue loss at most 1 from each of them.

443   To extend our result to $\mathcal{V} = [1, H]$, consider $i < v_t < i + 1$. Its cumulative utility of playing the
444   option $(i, j)$ with $\kappa_{i,j} < t \leq \kappa_{i,j+1}$ is

$$
\begin{aligned}
\sigma_{(i,j),t}(v_t) &= -\alpha \cdot \frac{\varepsilon T}{H} \cdot i + (G_i + \alpha) \cdot \frac{\varepsilon T}{H} \cdot v_t + (t - \kappa_{i,j}) \cdot (v_t - i) \\
&\geq G_i \cdot \frac{\varepsilon T}{H} \cdot v_t + \alpha \cdot \frac{\varepsilon T}{H} \cdot (v_t - i) \\
&= \sigma_{(i,j),t}(i) + (G_i + \alpha) \cdot \frac{\varepsilon T}{H} \cdot (v_t - i)
\end{aligned}
$$

445   The next lemma demonstrates that the buyer at round $t$ with valuation $i < v_t < i + 1$ and $t \in$
446   $(\kappa_{i,j}, \kappa_{i,j+1}]$, prefers the option $(i, j)$ over any option $(i', j')$ with $i' \notin \{i, i+1\}$.

447   **Lemma 11.** *For any option $(i', j')$ with $i' \neq i$, $\sigma_{(i,j),t}(v_t) - \sigma_{(i',j'),t}(v_t) > \gamma T$.*

448   *Proof.* Notice that its cumulative utility of playing the option $(i', j')$ with $i' < i$ is at most

$$
\begin{aligned}
\sigma_{(i',j'),t}(v_t) &= -\alpha \cdot \frac{\varepsilon T}{H} \cdot i' + (G_{i'} + \alpha) \cdot \frac{\varepsilon T}{H} \cdot v_t + (t - \kappa_{i',j'}) \cdot (v_t - i') \\
&\leq G_{i'} \cdot \frac{\varepsilon T}{H} \cdot v_t + \alpha \cdot \frac{\varepsilon T}{H} \cdot (v_t - i') + \frac{\varepsilon T}{3H^2} \cdot (v_t - i') \\
&= G_{i'} \cdot \frac{\varepsilon T}{H} \cdot v_t + \frac{\varepsilon T}{H} \cdot (v_t - i') \\
&= \max_t \{\sigma_{(i',j'),t}(i)\} + (G_{i'} + 1) \cdot \frac{\varepsilon T}{H} \cdot (v_t - i)
\end{aligned}
$$

449    Taking the difference between $\sigma_{(i,j),t}(v_t)$ and $\sigma_{(i',j'),t}(v_t)$ when $i' < i$, we have

$$\sigma_{(i,j),t}(v_t) - \sigma_{(i',j'),t}(v_t) \geq \left( \sum_{\tau=i'+1}^{i} (\frac{i}{\tau} - 1) \right) \cdot \frac{\varepsilon T}{H} + \left( -\frac{1}{3H} + \sum_{\tau=i'+1}^{i} \frac{1}{\tau} \right) \cdot \frac{\varepsilon T}{H} \cdot (v_t - i)$$

$$\geq 0 + \left( -\frac{1}{3H} + \frac{1}{i} \right) \cdot \frac{\varepsilon T}{H} \cdot (v_t - i) > \gamma T$$

450    where the first inequality partly follows the difference between $\sigma_{(i,j),t}(i)$ and $\max_t\{\sigma_{(i',j'),t}(i)\}$.

451    Next, notice that its cumulative utility of playing the option $(i', j')$ with $i' > i$ is at most

$$\sigma_{(i',j'),t}(v_t) = -\alpha \cdot \frac{\varepsilon T}{H} \cdot i' + (G_{i'} + \alpha) \cdot \frac{\varepsilon T}{H} \cdot v_t + (t - \kappa_{i',j'}) \cdot (v_t - i')$$

$$\leq G_{i'} \cdot \frac{\varepsilon T}{H} \cdot v_t + \alpha \cdot \frac{\varepsilon T}{H} \cdot (v_t - i')$$

452    Taking the difference between $\sigma_{(i,j),t}(v_t)$ and $\sigma_{(i',j'),t}(v_t)$ when $i' > i + 1$, we have

$$\sigma_{(i,j),t}(v_t) - \sigma_{(i',j'),t}(v_t) \geq (G_i - G_{i'}) \cdot \frac{\varepsilon T}{H} \cdot v_t + \alpha \cdot \frac{\varepsilon T}{H} \cdot (i' - i)$$

$$\geq (G_i - G_{i'}) \cdot \frac{\varepsilon T}{H} \cdot (i+1) + \alpha \cdot \frac{\varepsilon T}{H} \cdot (i' - i)$$

$$\geq \left( -\sum_{\tau=i+1}^{i'} \frac{i+1}{\tau} \right) \cdot \frac{\varepsilon T}{H} + \alpha \cdot \frac{\varepsilon T}{H} \cdot (i' - i)$$

$$= \left( -(i' - i) + \sum_{\tau=i+1}^{i'} \frac{\tau - i - 1}{\tau} \right) \cdot \frac{\varepsilon T}{H} + \alpha \cdot \frac{\varepsilon T}{H} \cdot (i' - i)$$

$$\geq \left( -(i' - i) + \frac{i' - i - 1}{i'} \right) \cdot \frac{\varepsilon T}{H} + \alpha \cdot \frac{\varepsilon T}{H} \cdot (i' - i)$$

$$= \left( -\frac{i' - i}{3H} - \frac{i+1}{i'} + 1 \right) \cdot \frac{\varepsilon T}{H}$$

453    The minimum is obtained when $i' = i + 2$ or $i' = H$. When $i' = i + 2$, we have

$$-\frac{i' - i}{3H} - \frac{i+1}{i'} + 1 = -\frac{2}{3H} - \frac{i+1}{i+2} + 1 = \frac{1}{i+2} - \frac{2}{3H} \geq \frac{1}{H} - \frac{2}{3H} > 0$$

454    while when $i' = H$, we have $i \leq H - 2$ and

$$-\frac{H - i}{3H} - \frac{i+1}{H} + 1 = -\frac{H + 2i + 3}{3H} + 1 \geq -\frac{H + 2(H-2) + 3}{3H} + 1 = \frac{1}{3H} > 0.$$

455                                                            □

456    Therefore, for $i < v_t < i + 1$ and $t \in (\kappa_{i,j}, \kappa_{i,j+1}]$, the buyer will play option $(i', j')$ with
457    $i' \notin \{i, i+1\}$ with probability at most $\gamma$.

458    Moreover, recall that it is indeed that $\kappa_{i,j} = \kappa_{i+1,j}$ for all $i$. Therefore, if the buyer plays option
459    $(i + 1, j)$, it will generate revenue $i + 1$. Moreover, if the buyer plays option $(i', j')$ with $j' < j$ and
460    $i' \in \{i, i+1\}$, then the option is already in its $\emptyset$ session and the buyer is going to pay $H$. Finally, for
461    any option $(i', j')$ with $j' > j$ and $i' \in \{i, i+1\}$, the utility of playing such an option is at most

$$\sigma_{(i',j'),t}(v_t) = -\alpha \cdot \frac{\varepsilon T}{H} \cdot i' + \left( t - \kappa_{i',j'} + (G_{i'} + \alpha) \cdot \frac{\varepsilon T}{H} \right) \cdot v_t$$

462    while the utility of playing the option $(i', j)$ is

$$\sigma_{(i',j),t}(v_t) = -\alpha \cdot \frac{\varepsilon T}{H} \cdot i' + (G_i + \alpha) \cdot \frac{\varepsilon T}{H} \cdot v_t + (t - \kappa_{i',j}) \cdot (v_t - i')$$

Taking the difference, we have

$$\sigma_{(i',j),t}(v_t) - \sigma_{(i',j'),t}(v_t) = (\kappa_{i',j'} - \kappa_{i',j}) \cdot v_t + (t - \kappa_{i',j}) \cdot (v_t - i')$$

Therefore, when $i' = i$, then $\sigma_{(i',j),t}(v_t) - \sigma_{(i',j'),t}(v_t)$ is clear positive. Moreover, when $i' = i + 1$, we have

$$
\begin{aligned}
\sigma_{(i+1,j),t}(v_t) - \sigma_{(i+1,j'),t}(v_t) &= (\kappa_{i+1,j'} - \kappa_{i+1,j}) \cdot v_t + (t - \kappa_{i+1,j}) \cdot (v_t - i - 1) \\
&\geq (\kappa_{i+1,j+1} - \kappa_{i+1,j}) \cdot v_t + (\kappa_{i+1,j+1} - \kappa_{i+1,j}) \cdot (v_t - i - 1) \\
&= (\kappa_{i+1,j+1} - \kappa_{i+1,j}) \cdot (2v_t - i - 1) \\
&> (\kappa_{i+1,j+1} - \kappa_{i+1,j}) \cdot (2i - i - 1) > \gamma T.
\end{aligned}
$$

Therefore, we finish showing that for $i < v_t < i + 1$ and $t \in (\kappa_{i,j}, \kappa_{i,j+1}]$, the buyer will play option $(i', j')$ with $i' \in \{i, i+1\}$ and $j' \leq j$ with probability at least $1 - K\gamma$. Thus, the revenue loss from $v_t$ is at most 1. Applying a similar argument as in Section 3.2, we can conclude that the expected revenue loss is $O(T)$.

## A.3 Proof of Theorem 7

We first prove a lower-bound for $\mathcal{V} = \{1, 2, \cdots, H\}$.

**Lemma 12.** *If the buyer with $\mathcal{V} = \{1, 2, \cdots, H\}$ is running a mean-based algorithm, a non-adaptive menu-based mechanism, which obtains expected revenue at least $\mathsf{Val} - O(\varepsilon T)$, must have $\Omega(\frac{H}{\varepsilon})$ options, when there exists a null option that always allocates and charges nothing. $\Omega(\frac{H}{\varepsilon})$ options are necessary even when the values of the buyer are drawn from an unknown distribution.*

*Proof.* Let $I_{i,t}(c)$ be a binary variable indicating whether the buyer with value $v_t = c$ plays the $i$-th option. Suppose there are $K$ options in total and let

$$P_i(c) = \sum_{t=1}^{T} \Pr[I_{i,t}(c) = 1] \cdot p_{i,t}$$

be the expected total revenue obtained from the $i$-th option when the buyer's valuations are $v_t = c$ for all $t$. Since the expected total revenue is at least $\mathsf{Val} - O(\varepsilon T)$, when the buyer's valuations are $v_t = 1$ for all $t$ in which the total expected revenue is at least $T - \mu \varepsilon T$ for some constant $\mu$, there must exist an option $i^*$ such that

$$P_{i^*}(1) \geq \frac{(1 - \mu\varepsilon)T}{K}.$$

Moreover, let $t^* = \sup\{t \mid \sigma_{i^*,t}(1) \geq -\gamma T\}$. $t^*$ is well-defined since $\sigma_{i^*,0}(1) = 0$. Notice that for all $t > t^*$, since the buyer is running a mean-based algorithm, we have $\Pr[I_{i^*,t}(1)] \leq \gamma$ due to the presence of the null option. Therefore, we have

$$\sum_{t \leq t^*} p_{i^*,t} + \sum_{t > t^*} \gamma \cdot p_{i^*,t} \geq P_{i^*}(1) \geq \frac{(1 - \mu\varepsilon)T}{K} \Rightarrow \sum_{t \leq t^*} p_{i^*,t} \geq \frac{(1 - \mu\varepsilon)T}{K} - \gamma H T.$$

where we use the fact that $0 \leq p_{i^*,t} \leq H$. Notice that the cumulative utility $\sigma_{i^*,t^*}(H)$ is

$$
\begin{aligned}
\sigma_{i^*,t^*}(H) &= \sum_{t \leq t^*} H \cdot a_{i^*,t} - p_{i^*,t} \\
&= H \cdot \sigma_{i^*,t^*}(1) + (H - 1) \sum_{t \leq t^*} p_{i^*,t} \\
&\geq \frac{(H-1)(1 - \mu\varepsilon)T}{K} - \gamma H^2 T
\end{aligned}
$$

Consider an environment when the buyer's valuations are $v_t = H$ for all $t$. Since the buyer is running a no-regret algorithm, her cumulative utility for the first $t^*$ rounds is at least $\sigma_{i^*,t^*}(H) - o(T)$. This is true because although the standard no-regret guarantee only applies to the final round $T$, the regret for the first $t$ rounds must also be $o(T)$, for any $t < T$. For the sake of contradiction, assume that the regret for the first $t$ rounds is $\Omega(T)$. Notice that the no-regret algorithm does not depend on the

future. Therefore, consider an environment where the rewards for all options after round $t$ are set to be 0, which results in a $\Omega(T)$ regret for the final round $T$. A contradiction.

In addition, notice that the revenue loss from the first $t^*$ rounds is at least the buyer's cumulative utility, and thus, the revenue loss is at least

$$\sigma_{i^*,t^*}(H) - o(T) = \frac{(H-1)T}{K} - O(\varepsilon T).$$

Finally, since the total revenue loss for $T$ rounds is at least the total revenue loss for the first $t^*$ rounds, in order to achieve $O(\varepsilon T)$ revenue loss, we must have $K = \Omega(\frac{H}{\varepsilon})$. $\qquad\square$

Moreover, observe that our proof only uses two sequences of valuations: a sequence with all 1 and a sequence of all $H$. Thus, our lower bound also applies to the stochastic settings with unknown distributions. Finally, Theorem 7 is a simple corollary of Lemma 12.

## B  Critical mechanisms

### B.1  Values from an unknown distribution

We will show that it is possible to achieve the approximation guarantees in Theorem 9 via a *first-price auction with decreasing reserve prices*. This is the same type of mechanism used in Braverman et al. [2018] to extract the full surplus from buyers drawn with a known distribution. Their construction requires complete knowledge of $\mathcal{D}$ in order to set these reserve prices over time (specifically, by solving a linear program whose coefficients depend on $\mathcal{D}$) – we show we can design a single construction that gets the same approximation factor to the total economic surplus without knowledge of $\mathcal{D}$.

In such a mechanism, at each time $t$ the seller specifies a reserve price $f(t)$, where $f$ is a decreasing function with range $[1, H]$. In each round, if the buyer bids above $f(t)$ they receive the item and pay $b$; otherwise, they do not receive the item and pay nothing. More formally, the allocation and payment rules $(a_t(b), p_t(b))$ are defined as follows: if $b \geq f(t)$, then $p_t(b) = b$, and $a_t(b) = 1$; otherwise, $p_t(b) = q_t(b) = 0$.

We will often find it easier to work with the function $x(v) : [1, H] \rightarrow [0, T]$ where $x(v) = 1 - \frac{1}{T} \cdot \min_{f(t) \leq v} t$. Note that $x(v)$ equals the number of rounds where a bidder with value $v$ has value higher than the reserve price $f(t)$ (in particular, $x(v)$ is an increasing function of $v$).

It can be shown (Braverman et al. [2018], Section C) that if the buyer is mean-based, the revenue obtained by the seller by using such an auction is given by

$$R(\mathcal{D}) = T \cdot \mathbb{E}_{v \sim \mathcal{D}} \left[ vx(v) - \max_w (v - w)x(w) \right] - o(T).$$

Since this is an expectation over $v$, we have the following lemma:

**Lemma 13.** *If the seller is using a first-price auction with decreasing reserve price, then $R(\mathcal{D})/\mathsf{Val}(\mathcal{D})$ is maximized when $\mathcal{D}$ is a singleton distribution.*

We now proceed to prove Theorem 9.

*Proof of Theorem 9.* Fix any constant $0 < \varepsilon < 1$ (e.g. $\varepsilon = 1/2$). The seller will use a first price auction with descending reserve price given by

$$f(t) = \max \left( \exp \left( \frac{1}{C} \cdot (1 - \eta - \frac{t}{T}) \right), 1 \right),$$

where $\eta = (1 + \log H)^{-\varepsilon}$ and $C = \frac{1-\eta}{1+\log H}$. Note that for this choice of $f$, $x(v) = \eta + C \log v$. By Lemma 13, $R(\mathcal{D})/\mathsf{Val}(\mathcal{D})$ is maximized when $\mathcal{D}$ is a singleton distribution. We therefore have that:

$$\frac{R(\mathcal{D})}{\mathsf{Val}(\mathcal{D})T} \geq \min_v \frac{vx(v) - \max_w (v-w)x(w)}{v}$$

$$= \min_{v,w<v} \left( x(v) - \left(1 - \frac{w}{v}\right) x(w) \right)$$

$$= \min_{v,w<v} \left( (\eta + C\log v) - \left(1 - \frac{w}{v}\right)(\eta + C\log w) \right)$$

$$= \min_{v,w<v} \left( C\log \frac{v}{w} + \frac{w}{v}(\eta + C\log w) \right).$$

For a fixed $w$, this is minimized when $v = w\left(\frac{\eta}{C} + \log w\right)$. It follows that

$$\frac{R(\mathcal{D})}{\mathsf{Val}(\mathcal{D})T} \geq \min_w \left( C\log \left(\frac{\eta}{C} + \log w\right) + 1 \right)$$

$$\geq C\log \left(\frac{\eta}{C}\right)$$

$$\geq \frac{(1 - (1 + \log H)^{-\varepsilon}) \log((1 + \log H)^{1-\varepsilon})}{1 + \log H}$$

$$= \Theta \left( \frac{\log \log H}{\log H} \right).$$

□

## B.2   Adversarial values

In this section we will show that it is possible to achieve a $(1/\log H)$-approximation to the buyer's welfare if they are playing a mean-based algorithm with adversarial values supported on $[1, H]$. This will follow naturally from the known fact that it is possible to achieve the same approximation guarantee against a *strategic* buyer with value in $[1, H]$ playing a single-round version of this game (see e.g. Chapter 6 of Hartline [2013]) – we simply show that if we run this mechanism every round, mean-based buyers will learn to bid in the same manner as strategic buyers.

Our mechanism (equivalent to the mechanism presented in Theorem 6.5 of Hartline [2013]) is as follows: for each $b$ and for all $t$, we set the allocation probability $a_t(b) = (1 + \log b)/(1 + \log H)$ and the expected price charged to $p_t(b) = b/(1 + \log H)$. Note that this mechanism can also be interpreted as a second-price auction where the buyer draws a random reserve from the distribution with CDF $F(r) = \frac{1 + \log r}{1 + \log H}$. It can be seen that for any $v \in [1, H]$, the expected utility $U(v, b) = v \cdot a_t(b) - p_t(b)$ of bidding $b$ with value $v$, is maximized when $b = v$. A strategic buyer therefore will always bid their value, and pay $1/(1 + \log H)$ of their value in total.

We will show that the mean-based guarantee ensures that a mean-based buyer will (most of the time) choose a bid close to $v$, and thus contribute a similar amount of revenue as a strategic buyer.

**Lemma 14.** *Let*

$$U(v, b) = v \cdot a_t(b) - p_t(b) = \frac{v \cdot (1 + \log b) - b}{1 + \log H}.$$

*If $U(v, v) - U(v, b) \leq \delta$, then $|v - b| \leq H\sqrt{2(1 + \log H)\delta}$.*

*Proof.* Let $f(v, b) = v(1 + \log b) - b$. Note that

$$\frac{\partial}{\partial b} f(v, b) = \frac{v}{b} - 1$$

and

$$\frac{\partial^2}{\partial b^2} f(v, b) = -\frac{v}{b^2} \leq -\frac{1}{H^2}.$$

This implies $f$ is $(1/2H^2)$-strongly concave and maximized when $b = v$, so

$$f(v, v) - f(v, b) \geq \frac{1}{2H^2}(v - b)^2.$$

Since $U(v, b) = f(v, b)/(1 + \log H)$, this implies that if $U(v, v) - U(v, b) \leq \delta$, then $|v - b| \leq H\sqrt{2(1 + \log H)\delta}$. $\qquad\qquad\square$

*Proof of Theorem 10.* Consider the critical mechanism defined by $a_t(b) = (1 + \log b)/(1 + \log H)$ and $p_t(b) = b/(1 + \log H)$. We claim that this mechanism obtains expected revenue at least $\frac{1}{1+\log(H)}\mathsf{Val} - o(T)$ against any mean-based bidder with values adversarially chosen from $[1, H]$.

Note that by the mean-based guarantee, with probability at least $1 - \gamma$ a mean-based algorithm will pick a bid $b_t$ satisfying

$$\sigma_{v_t, t}(b^*) - \sigma_{v_t, t}(b_t) \leq \gamma T, \tag{2}$$

where $b^* = \operatorname{argmax}_b \sigma_{v_t, t}(b)$. Now, note that $\sigma_{v, t}(b) = t \cdot U(v, b)$; since $U(v, b)$ is maximized when $b = v$, $b^* = v_t$. It then follows from Lemma 14, that if (2) holds, then with probability at least $(1 - \gamma)$

$$|v_t - b_t| \leq H\sqrt{2(1 + \log H)\frac{\gamma T}{t}}. \tag{3}$$

Since $v_t - b_t \leq H$ is always true (since $v_t \leq H$), it follows that in expectation,

$$\mathbb{E}[b_t] \geq v_t - \gamma H - H\sqrt{2(1 + \log H)\frac{\gamma T}{t}}$$

and therefore we have that

$$
\begin{aligned}
\mathbb{E}[\mathsf{Rev}] &= \mathbb{E}\left[\sum_{t=1}^{T} p_t(b_t)\right] \\
&= \frac{1}{1 + \log H}\sum_{t=1}^{T}\mathbb{E}[b_t] \\
&\geq \frac{1}{1 + \log H}\sum_{t=1}^{T}\left(v_t - \gamma H - H\sqrt{2(1 + \log H)\frac{\gamma T}{t}}\right) \\
&= \frac{\mathsf{Val}}{1 + \log H} - \frac{H}{1 + \log H}\gamma T - H\sqrt{\frac{2}{1 + \log H}\gamma T}\sum_{t=1}^{T}\frac{1}{t} \\
&\geq \frac{\mathsf{Val}}{1 + \log H} - \frac{H}{1 + \log H}\gamma T - H\sqrt{\frac{2}{1 + \log H}}\gamma T \\
&= \frac{\mathsf{Val}}{1 + \log H} - o(T),
\end{aligned}
$$

where the last inequality holds since $\gamma$ and $\sqrt{\gamma}$ are both $o(1)$.

$\qquad\qquad\square$