[Reviews · NeurIPS 2019]

Reviewer 1



The authors investigate several settings of selling an item to a single buyer repeatedly over T timesteps, with the goal to maximize revenue. The assumption is that the buyer uses a no-regret algorithm to decide which option (that the seller offers) to choose (given the private value that they have). The authors propose 1-eps approximations of the entire surplus when values are adversarially chosen using a polynomial (in eps^-1) number of options. When restricting attention to the class of critical mechanisms, they achieve revenue guarantee of O(log log H / log H) for stochastic values and O(1/log H) for adversarial values (where H is the support of the distribution). While the problem is well-motivated, the exposition seems pretty informal at times which makes the claims difficult to verify (see details under “improvements”).

Reviewer 2



This work builds significantly on the results from [Braverman et al 2018] by reducing dependence on the form and structure of buyers value distributions, very much in line with the prior-free and prior-independent thread in AGT. The paper first considers menus of prices, and shows in a rather unrealistic mechanism that the full welfare can be extracted, without knowing the distribution exactly, just the support, and that the number of options is essentially tight. [Braverman et al 2018] gave a similar algorithm, but with dependence on the distributions and more importantly the existence of a distribution in each round. These approaches rely very heavily on the mean-based no regret learning, as there are easy improvements in policies if the learner isn't restricted to mean based. Focusing on mechanisms that are likely to be used, they give a decreasing reserve price for use in a first price auction. [Braverman et al 2018] had given a decreasing reserve price according to an LP that depended explicitly on the value distribution; this work provides a decreasing reserve price just based on the support of the distribution. As it is building directly on Braverman et al 2018, it is not introducing any significant new models, but it is making the approach significantly more robust, as well as illustrates an approach for eliminating prior dependence that may be interesting in other dynamic settings. The paper is generally very well written.

Reviewer 3



This study deals with pricing mechanisms against a no-regret buyer in prior-independent setting and in prior-free one. In the setup, a single seller interacts with a single buyer. The authors investigate the case of contextual auctions as well. The paper is well written and well structured. The submission is technically sound. However, there are weaknesses as follows: 1. Not clear positioning with respect to related work. The authors write in Lines 49-52: “This opens a wealth of questions of how to robustly design dynamic mechanisms that perform well in the worst-case against some class of learning agents. In this paper, we explore this question for one of the simplest problems in dynamic mechanism design: repeatedly selling a single item to a single buyer for T rounds.” But, there is a large body of studies devoted to worst-case scenario in repeated auctions between a single seller and a single buyer in ML venues: (Amin et al, NIPS’2013), (Mohri&Medina, NIPS’2014), (Drutsa, ICML’2018), (Huang et al, NeurIPS’2018), etc. None of the above has been cited by the authors. [after author feedback: OK, the answer is clear for me. Please, put it in the next version of your paper. The main reason of ambiguity here consists in the following: past few years a very large variety of papers on dynamic auctions/mechanisms has been published. I would suggest authors to highlight the key parameters of a setup in the field of dynamic auctions/mechanisms (make a scheme), e.g.: - single buyer or multiple; - myopic buyer/ strategic one / learning buyer/ etc; - prior-free, prior-[in]dependend, - etc. For instance, it would be great to use a table if space permits. Then, you put related work in this scheme and put your study here.] 2. Why the no-regret buyer is modeled via Bandits algorithm? [after author feedback: I did not get an answer for this; and I do not understand it still. I confirm that Nekipelov et al. argue that the setting is realistic. However, the question was: why a no-regret algorithm is exactly a Bandit algorithm? For instance, the series of works (Amin et al, NIPS’2013), (Mohri&Medina, NIPS’2014), (Drutsa, ICML’2018) consider no-regret algorithms, but they are not Bandit ones.] Do the results hold for a buyer from the study [Heidari et al, ICML’2016]? [after author feedback: OK, the answer is clear for me. Please, put it in the next version of your paper.] 3. Most proofs of theoretical statements are moved to Supplementary Materials. I think, some of the key statements should be supported by at least a sketch proof in the main test. [after author feedback: OK, but I believe that key/central statements should be supplied by key intuitions behind their proofs even in a hard space limit. If the work is so large that it cannot match this, it may be a signal that the paper is better suited for other venue with larger format.] 4. Unclear text in Lines 68-70: “If the option-complexity of the mechanism begins to scale with the number of rounds T, this may even nullify any sort of low-regret or mean-based guarantee the learning algorithm has (it may not even be possible to explore all potential options).” [after author feedback: OK, I get it.]

Reviewer 4



Originality: reasonable. The paper is a follow-up on Braverman et al. 2018, but it extends that model to adversarial arrivals. I believe the main algorithmic contribution is nicely original, though the approach is probably not that surprising to experts in hindsight. Quality: Good in my opinion. I didn't find errors. Although, the math is very terse in the main submission, I didn't verify proofs. Clarity: Very good in my opinion. My only complaint was not having a helfpul high-level description of the algorithms. Significance: Medium. The paper addresses an extension of the problem that seems very natural. The main significant contribution seems to be the option-based algorithm, but the other results also seem helpful in understanding the space. Other comments: I appreciated that Section 3 goes from easy cases to more difficult ones, but I still had trouble pulling the key idea from even the "warm-up" -- some more intuition would have been helpful. I was a bit confused whether the theorems really capture "extracting full surplus". Any instantiation of the algorithm loses O(T). We can apparently make the constant factor as small as we want, but it's still qualitatively different than an o(T) guarantee.

[Author Response · NeurIPS 2019]

We thank all the reviewers for their insightful reviews. We address the comments of each reviewer separately.

**Reviewer #3:** Regarding your comment 'the authors claim "the buyer does not play..." ': as we state in line 211, the
goal of that paragraph was to give a high level idea of the construction. Indeed, while elaborating the details later on,
we make precise this very point in line 233: there we say that the buyer's probability of playing the 0-th option (which
we also call as the high option, as mentioned in line 212) is at most $\gamma$. Since $\gamma = o(1)$ for mean-based algorithms, we
said in line 214 (the high level paragraph) "the buyer does not play"; indeed, at a high level, playing with probability
$o(1)$ is equivalent to not playing at all, and in fact it does not affect the asymptotics of our regret guarantee whether
the arm is played with probability $o(1)$ or 0 (as the difference would amount to at most $o(T)$ additive regret). We will
ensure that we leave no room for ambiguity even in the high level ideas section in the next version.

The previous paragraph also answers your questions "And how does one guarantee that an arm is not played? In
most MAB algorithms, aren't even historically unhelpful arms are played with some low probability?". Indeed, even
historically unhelpful arms are played with some low probability, and in our case this low probability is $\gamma = o(1)$, as
mentioned in line 233.

Regarding your comment "I was under the impression that the authors assumed the buyer was running a multi-arm
bandit algorithm, which doesn't observe all rewards at each time step": yes, the buyer is indeed running a multi-armed
bandit algorithm, and, the buyer can only observe the reward of the option that he chose, not of all the options.

Regarding your comment of "utility gain from session 0 is exactly cancelled out with his utility loss from session
2 (line 236)" and the related question of "How then can you guarantee that the observed utilities exactly sum up to
some constant?": we believe the ambiguity gets cleared by focusing on the word "observed" in your comment. What
we meant by "cancelled out" is that the cumulative utility of playing arm $i$ (*not the cumulative observed utility*) is 0.
Formally, $\sigma_{i,t}(1) = 0$ for any $t > \kappa_{i+1}$ where $\sigma_{i,t}(1)$ (formally defined in Definition 2 line 185) is the cumulative
utility of playing option $i$ for all of the first $t$ rounds with a buyer value of 1. Clearly, in a multi-armed bandit setting
the cumulative utility for playing an arm $i$ (not cumulative observed utility), can be set by the adversary/seller in an
arbitrary manner to whatever the adversary/seller wants, regardless of the buyer's strategy. The interesting question then
is how can the cumulative utility (instead of cumulative observed utility) have any consequence on the probability with
which the player/buyer plays an arm/option? This is where the definition of a mean-based strategy helps (Definition 2,
line 185): it has strong consequences on the probability with which a player/buyer plays an arm/option based purely on
cumulative utilities. While this may seem very restrictive at first glance, surprisingly several common/natural no-regret
algorithms are mean-based: for instance the famous EXP3 algorithm for the multi-armed bandit setting is a mean-based
algorithm. We will make sure to emphasize this and make it fully clear.

Given that these were Reviewer 3's main concerns, we sincerely hope Reviewer 3 would consider revising their score.

**Reviewer #4:** We thank the reviewer for the positive and encouraging review. It is a very interesting and challenging
open question to extend both our results and Braverman et al.'s results to the multi-agent setting.

**Reviewer #5:**

1. Regarding the collection of related works that you mention: in all these works, the *seller is learning* to set prices
for a strategic buyer. Whereas in our paper the *buyer is learning* a good bidding strategy using a no-regret learning
algorithm. So our work is fundamentally different from all these related works both in subject and in techniques. We
will add a short discussion to emphasize this crucial difference from the related works you mention.

2. Regarding the comparison to Heidari et al.: Heidari et al. [ICML16] studies a setting in which the buyers use no-regret
algorithms for choosing ad exchanges. In our paper there is only one ad exchange (i.e., one seller), and the buyers use
no-regret algorithms to learn a bidding strategy. We believe that our setting is quite realistic: in fact Nekipelov et al.
[EC15] (cited in line 112 in our paper) provide a theoretical model to empirically confirm that advertisers (i.e., buyers)
indeed behave as if they are running no-regret algorithms to learn their bidding strategies.

3. Regarding proofs being moved to Appendix: we would have loved to discuss more in the main body. Given the
technical nature of our proofs and the 8-page limit, we had to move most proofs to Appendix. We will make sure to
give proof sketches in the next version.

4. Regarding the text in line 68-70: what we mean here is that if the number of options $K$ we provide to the buyer is
too large or infinite, then the no-regret guarantee for bandit algorithms would become very weak. This is because the
regret increases with $K$ (apart from increasing with $T$): for example, the regret bound for the EXP3 bandits algorithm
is $O(\sqrt{KT})$; if $K$ gets quite large, say $K = \Theta(T)$, this regret bound becomes trivial/vacuous (i.e., linear in $T$) like we
mention in lines 146-147. Thus it is important to strive to design mechanisms that have a small $K$ (at the very least, $K$
which grows sublinearly in $T$), and this is something we focus on in our paper.

Given that these were Reviewer 5's main concerns, we sincerely hope Reviewer 5 would consider revising their score.

[Meta-Review · NeurIPS 2019]

The paper considers "selling to a no-regret buyer": repeatedly selling to a buyer that learns over time using a no-regret algorithm. It follows up on (Braverman et al, Best Paper at ACM EC 2018), and obtains several interesting improvements. While the reviewers largely agree on the significance of the contributions, some of the reviewers are very concerned with presentation. While they agree all their concerns can be addressed in a revision, they insist these concerns *must* be addressed if the paper is accepted. While there is a large spread in the scores, there was a thorough discussion among the reviewers and the AC. This paper is closely related to Submission #908, with some overlap in the co-authors, but is sufficiently different. This relation should be discussed carefully. As per one of the comments, the claims re extracting full surplus may need to calibrated a little more carefully.